# Using MODIS-NDVI Time Series to Quantify the Vegetation Responses to River Hydro-Geomorphology in the Wandering River Floodplain in an Arid Region

Xarapat Ablat [1], Gaohuan Liu [2,*], Qingsheng Liu [2] and Chong Huang [2,*]

1  Department of Water Resources and Environment, School of Geography and Planning,
   Sun Yat-sen University, Guangzhou 510275, China; abulait@mail.sysu.edu.cn
2  State Key Laboratory of Resources and Environmental Information System, Institute of Geographic Sciences
   and Natural Resources Research, Chinese Academy of Sciences, Beijing 100101, China; liuqs@lreis.ac.cn
*  Correspondence: liugh@lreis.ac.cn (G.L.); huangch@lreis.ac.cn (C.H.)

**Abstract:** Vegetation, hydrology and geomorphology are three major elements of the floodplain ecosystem on Earth. Although the normalized difference vegetation index (NDVI) has been used extensively to characterize floodplain vegetation growth, vigour and biomass, methods for quantifying the various distinct responses of floodplain vegetation to hydro-geomorphological changes in different lateral belts in arid regions are still needed. In this study, the Linhe reach was divided into four lateral belts based on their hydro-geomorphological characteristics, and the Moderate Resolution Imaging Spectroradiometer (MODIS)-NDVI time series statistical indicators were used to characterise the distinct changing the patterns of vegetation growth in different belts. The response of floodplain vegetation to river hydro-geomorphology in each belt was analysed. The result showed that the average maximum NDVI value in the regular inundation area was 0.23 and higher than that in the other lateral belts. The correlation between the water persistence time and peak NDVI value in the regular water inundation area was significant ($\rho = 0.84$), indicating that in contrast to highly frequent or extremely rare water inundation, regular water inundation provides significant benefits to floodplains. Continuous or highly frequent inundation may cause decreased vegetation productivity. Overall, our results suggest that the vegetation greenness response to the river hydro-geomorphology is different from the river to the edge of the floodplain. Thus, a better understanding of the interactions between the floodplain vegetation and river hydro-morphology and river water resource management in arid-region floodplains.

**Keywords:** riparian vegetation; fluvial processes; hydro-geomorphological dynamics; MODIS-NDVI time-series; the upper Yellow River; dry regions

## 1. Introduction

Vegetation, hydrology and geomorphology are the three main controlling elements of floodplain ecosystems on Earth, especially in dry areas [1–4]. Differences in landscape shapes significantly rely on the amount of water and the geomorphic processes that affect the runoff in the basin [5,6]. Investigating the response of vegetation to hydro-geomorphological processes can provide insight into floodplain changes and lead to improved management of floodplain ecosystems [7].

The investigation of the influences of hydro-geomorphology on fluvial vegetation began in the late decades of the 20th century. Many scholars recognised that the influence of hydro-geomorphological distribution on vegetation in the floodplain areas resulted in significant spatial distribution patterns of vegetation composition and structure, which was related to the riverbed and the geomorphological characteristics of the floodplain area and its evolution [6,8–19]. In addition, some studies revealed that changes in floodplain vegetation can, in turn, influence hydro-geomorphological processes [11,20–24].

Spaceborne remote sensing technology is widely used to quantitatively study the response of vegetation growth and biomass to hydrological and geomorphic factors in floodplain areas [25–27]. Furthermore, remote sensing data are considered one of the main data types for characterizing aspects of large floodplains in dryland environments where there are few cloudy days and a low density of vegetation [28]. Readily available Landsat imagery with a long record provides sufficient spatial resolution to capture subtle dynamics of vegetation at the landscape scale but its course time resolution and interference of clouds make the acquisition of a detailed time-series of vegetation greenness over a short time period unlikely [17,29]. The two Sentinel-2 satellites that launched in recent years, like Landsat, are freely available worldwide, and also have a high spatial resolution (10–60 m). Although the spatial resolution is highly consistent with Landsat, the revisit time is reduced to 5 days. MODIS (Moderate Resolution Imaging Spectroradiometer) images which 1-day frequency can reflect the change of vegetation greenness in a highly dynamic riverbed, but their coarse spatial resolution provides a sub-optimal representation of the instantaneous distribution of vegetation greenness, especially in flat landscapes with complex riverbed networks [30–33]. The NDVI is widely used all over the world for characterizing vegetation greenness, vigour and biomass [34]. NDVI is the most commonly used vegetation index, which compares the reflectance intensity of the red band and near-infrared band to quantify green living vegetation [35]. Through the band intensity ratio, NDVI removes a large amount of noise caused by cloud shadow, terrain and solar angle changes, as well as atmospheric attenuation in visible and infrared bands, which makes NDVI less sensitive to lighting conditions [36–42]. In recent decades, time-series NDVI products with different spatial resolutions have become available, enabling investigations of ecosystem changes and large-scale land monitoring/mapping to detect changes associated with human activities [43,44].

Despite significant research progress, little research has aimed at quantifying the changes in the spatial distribution of vegetation in response to river hydrogeomorphology along a gradient from the riverbed to the edge of floodplains based on remote sensing observations, especially in high dynamics wandering river floodplains. The objective of this study was to quantify the spatial changes of vegetation in responses to river hydro-geomorphology in a floodplain region using NDVI- associated indicators derived from a time-series of daily MODIS-NDVI images. To this end, based on the floodplain hydro-geomorphological characteristics, the study area was divided into four lateral belts spanning from the riverbed to the floodplain margins to investigate the vegetation responses under the complex floodplain hydro-geomorphological processes. Multiple indicators derived from a pixel-based NDVI time-series, including peak NDVI values, frequency of NDVI values, and water inundation time were used to characterise the spatial distribution of vegetation vigour and biomass and analyse the relationships between the different vegetation distribution patterns and floodplain hydro-geomorphology.

## 2. Materials and Methods

### 2.1. Study Area

The study was performed along the most northern part of the Yellow River, which is located in the southwestern Inner Mongolia of China (107°24′ E, 40°38′ N–107°26′ E, 40°39′ N) (Figure 1A). The study region has a typical semi-arid continental climate with four distinct seasons, which are characterised by less precipitation and more evaporation. The yearly mean precipitation for the study area was 150–400 mm from 2010 to 2015, and three-quarters of the annual rainfall is concentrated in summer (between June and September). During this period, the mean annual air temperature was 8.2 °C, and the regions experience 135–150 frost-free days and 3100–3300 h of sunshine per year [45]. The area comprises wandering river floodplains with a total area of 150 km$^2$ and a length of 30 km, and the widest part of the floodplain reaches 5 km. The study area experiences strong geomorphological dynamics and exhibits a very gentle slope of 1/10,000 (Figure 1B–D) [46]. During the study period (2010–2015), the interannual hydrological regime has been typi-

cally stable. Summer water levels slightly increased between July and October each year. Because of the geographical conditions, the channel depth has decreased, and flows from low to high latitudes together have caused ice flooding phenomena each year [47]. In winter, the downstream areas freeze earlier than does the upstream portion of the study reach, which causes the water level to rapidly increase, Thus, the river water in the study reach experiences a "banking" phenomenon each winter [48]. The floodplain area is inundated with water during this period, and low-temperature conditions cause inundated waters to quickly freeze in floodplain areas rather than re-entering riverbeds.

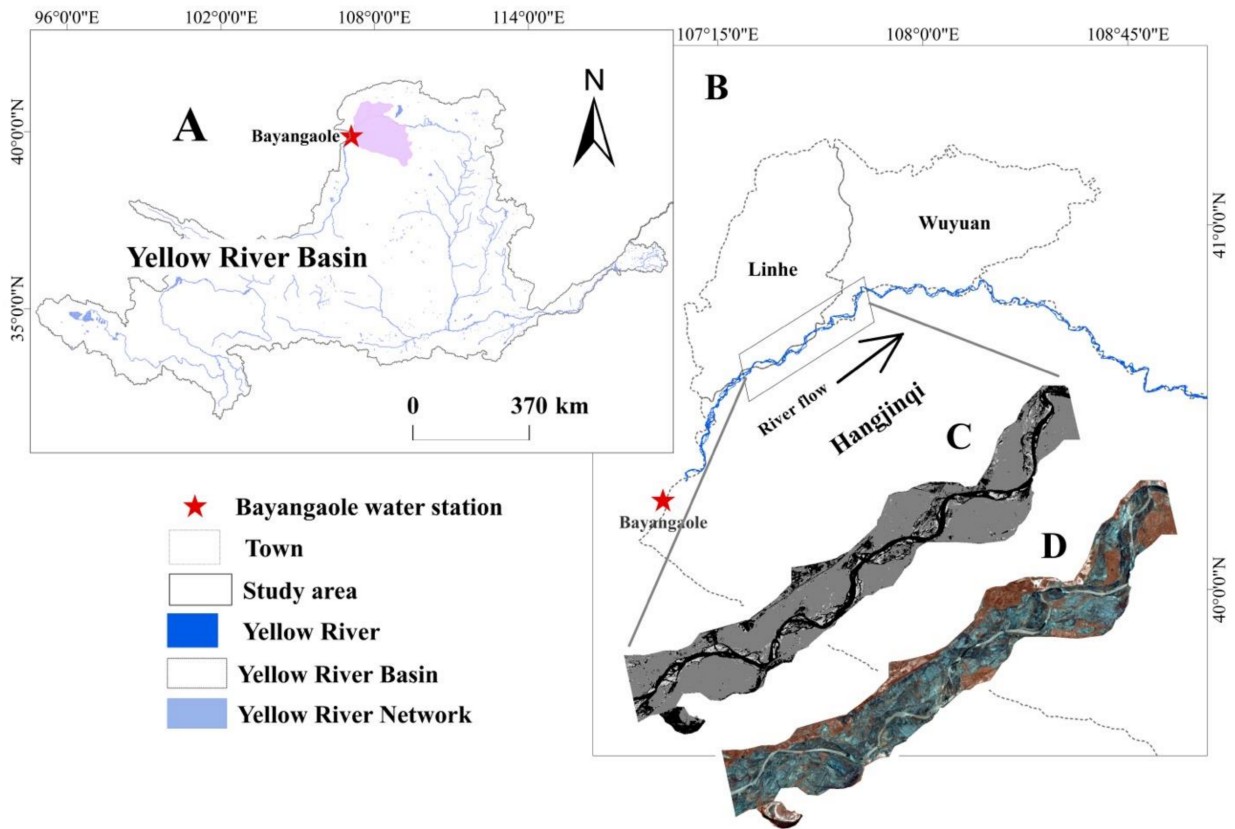

**Figure 1.** Location of the study area and conceptual model of hydrogeomorphological interaction patterns (**A**) The Yellow River Basin and location of the Linhe Reach (indicated by the solid line box); (**B**) The Linhe Reach and location of the study area (indicated by the dashed-line box); (**C**) NDVI image of the study area on the 15 August 2015; Pixel-based points P1–P4 are located along a cross-section at different distances from riverbed; (**D**) False colour composite Landsat 8 of the study area collect from 12 January 2015.

The natural vegetation in the study area is mainly herbaceous. The dominant vegetation type is salt-tolerant vegetation, including species such as *Cynodon dactylon*, *Suaeda salsa (L.) Pall* and *Sporobolus virginicus (L.) Kunth*. Vegetation growth is mainly reliant on the persistence of "banked up" water caused by ice jam flooding every winter [12].

### 2.2. Datasets

Remote sensing images (Landsat and MODIS) and hydrological measurements obtained from the Bayangaole water station were used in our study.

Landsat OLI (Landsat Operational Land Imager) images with a spatial resolution of 30 m were used to divide the study area into lateral zonal belts based on the hydrological dynamic characteristics. The images were obtained from the Aerospace Information Research Institute (AIR) of Chinese Academy of Sciences (http://ids.ceode.ac.cn/). The images corresponded to three dates: 14 August 2014, 24 April 2014 and 10 March 2015

which were based on the intra-annual hydrological dynamics of the study reach. The near-infrared, red and green bands were stacked for further classifications.

NDVI data were obtained from 8000 Daily Aqua/Terra MODIS surface reflectance data (MODIS9GQ) images (horizontal number 26 and vertical number 4, 250 m) covering the period 1 January 2010 to 31 December 2015 downloaded from Atmosphere Archive and Distribution System Distributed Active Archive Center (LAADS DAAS) at the Goddard Space Flight Center website (https://labsweb.modaps.eosdis.nasa.gov). The MODIS-NDVI images were subjected to standard atmospheric corrections [49] and re-projected from the sinusoidal to the WGS84 geographic coordinate system.

The average daily water level and discharge data used in this study were collected at the Bayangaole water station located in the upstream region of the study area (40°19′ N, 107°02′ S) (Figure 1), and were obtained from the Yellow River Conservancy Commission (YRCC).

### 2.3. Lateral Zonal Distributions of the Study Area

The study area is a typical wandering river reach of the Yellow River, and the geomorphological dynamics along the river is mainly driven by the river hydrological changes. In this paper, based on the high dynamic hydro-geomorphological characteristics of the study area, the Index of Hydrological Alteration (IHA) method [50] and Normalized Difference Water Index (NDWI) were used, and the entire study reach was divided into four belts from the river to the edge of the floodplains (Table 1). The detailed steps are as follows: Frist of all, according to the IHA method, the daily runoff data during 2008–2014 were used to determine the high and low flow thresholds, and after which average daily flows above 731 $m^3$/s in 2014 were distinguished as high flows. The flow frequency per 100 flows was calculated, it is known that the flow frequency between 1000–1100 $m^3$/s is the highest, which occurs 27 times within the year. Therefore, the Landsat remote sensing image that corresponds to the flow occurrence date in 2014 was applied to divide the whole study area into a highly dynamic area and relatively static area. The surface water area extracted from a Landsat 8 OLI image on 14 August 2014 (with an average daily flow of 1020 $m^3$/s) used NDWI (>0.4) based on the ENVI5.0 software represents the highly dynamic study area within a year; the rest of the study area is a relatively static area. The highly dynamic area included the continuously inundated areas and the frequently inundated floodplain areas. The extremely low flows (≤200 $m^3$/s) for 2014 were determined using the IHA method, and the inundation area of extremely low flows can be represented by the continuously inundated areas during this year. Therefore, the Landsat image was used to classify the highly dynamic area into a continuously inundated area and a frequently inundated floodplain area. The surface water area extracted from a Landsat 8 OLI image from 24 April 2014 (average daily flow of 209 $m^3$/s) used NDWI (>0.33) based on the ENVI5.0 software represents the continuously inundated study area within a year; the rest of the highly dynamic area is a frequently inundated area. According to the winter flooding inundation phenomenon in the study reach, was used to divide the relatively static area into a regularly inundated floodplain area and an extremely rarely inundated floodplain area. The ice cover area extracted from an image 10 March 2015 (the day on which the winter flooding inundated area reached the seasonal maximum) used NDWI (>0.27) based on the ENVI5.0 software represents the regular inundation area of the study area, the rest of the relatively static area is an extremely rarely inundated area. This border corresponded to an embankment road between the old floodplain area and the newly formed floodplain area, which created an approximately 1 m elevation difference. Therefore, the entire study area from the river to the floodplain margins was divided into four zones, including a continuously inundated area (belt 1), a frequently inundated floodplain area (belt 2), a regularly inundated floodplain area (belt 3) and an extremely rarely inundated floodplain area (belt 4).

**Table 1.** Water levels and discharges for the three analysed Landsat images.

| Remote Sensing Images | Spatial Resolution | Data | Water Level/m | Discharge/m$^3$/s |
|---|---|---|---|---|
| Landsat OLI | 30 m | 2014.08.14 | 1051.34 | 1020 |
| Landsat OLI | 30 m | 2014.04.24 | 1050.45 | 253 |
| Landsat OLI | 30 m | 2015.03.10 | 1049.31 | 142 |

The steps and results of the zonal distribution of the floodplain based on its hydro-geomorphological characteristics [17] are shown in Figure 2.

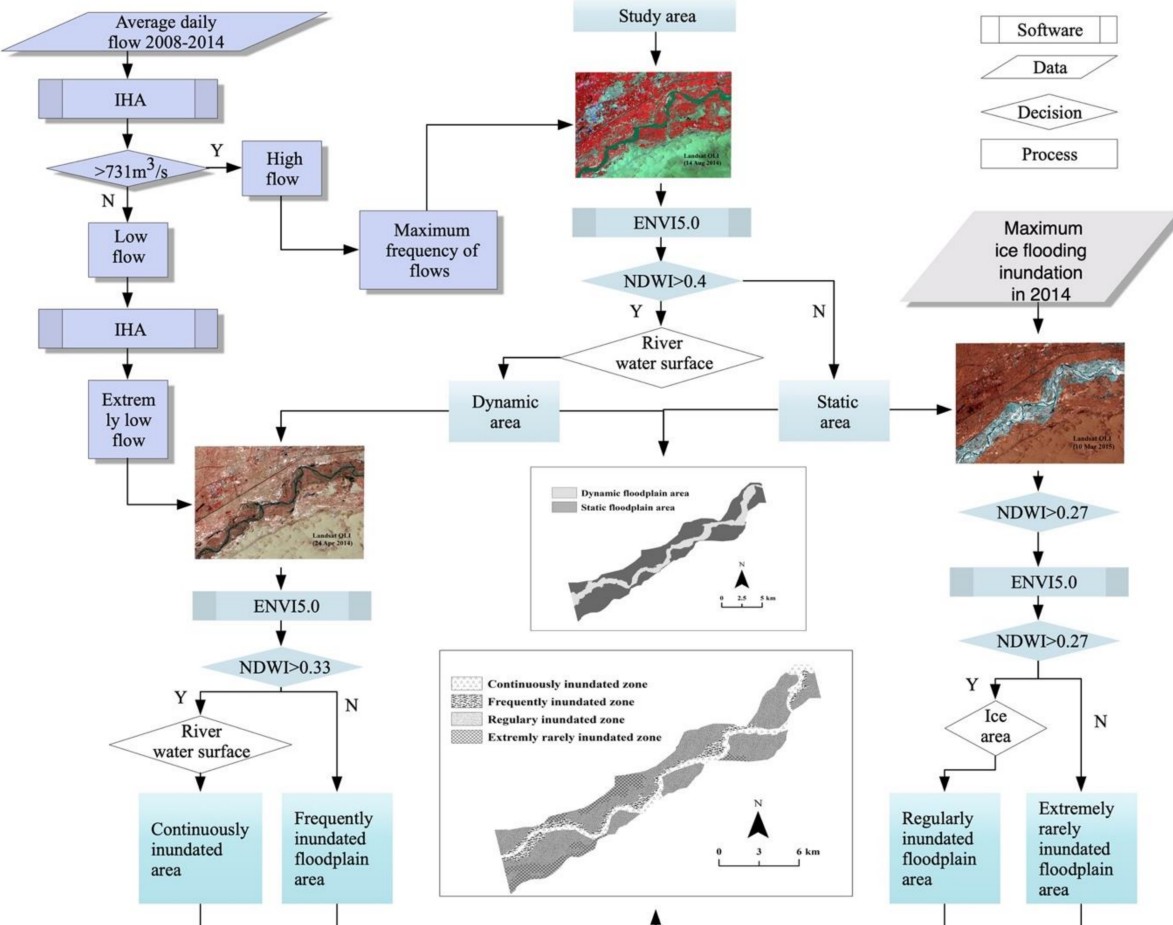

**Figure 2.** Workflow determining the four floodplain zones and the areas corresponding to different levels of hydro-geomorphological activity based on the floodplain hydro-geomorphology in the study years. The purple colour represents the process of the river hydrology, the light blue colour represents the zonal division process of the study area, dark blue represents the final results.

### 2.4. MODIS-NDVI Data Preprocessing

NDVI has been widely used for detecting vegetation productivity in arid areas around the world [51]. In the present study, NDVI was used to study the response of vegetation to the floodplain hydrogeomorphology, using the following equation:

$$\text{NDVI} = \frac{(B_2 - B_1)}{(B_2 + B_1)} \quad \text{where} \begin{cases} \text{the MODIS 9GQ 2} - \text{band } (841 - 876 \text{ nm}) \\ \text{the MODIS 9GQ 1} - \text{band } (620 - 670 \text{ nm}) \end{cases} \tag{1}$$

NDVI values range between $-1$ and 1; a value of $-1$ corresponds to a complete lack of vegetation, such as an open water area, and a value of 1 corresponds to an area fully covered

by green vegetation [51]. Here, NDVI-based indicators, including maximum NDVI, frequencies of individual NDVI values and water submergence period, were used to quantify the influences of the hydro geomorphological characteristics on the floodplain vegetation.

The maximum NDVI value was determined as the highest NDVI value of each pixel across all periods.

The following equation was used:

$$P_{NM} = \text{Maximum} \left( P_{t1(NDVI)}, P_{t2(NDVI)}, P_{t3(NDVI)}, \ldots, P_{tn(NDVI)} \right) \tag{2}$$

where $P_{NM}$ is the maximum NDVI value of $pixel_{(x,y)}$ in the images, and t and n are the collection time and the number of images, respectively. $P_{NM}$ values were calculated from daily time-series NDVI images between 1 April 2015 and 31 October 2015 using ENVI software and it was used to characterise floodplain zones from the in-channel region to the edge of the floodplains in the study area.

The frequency of an NDVI value was calculated as the number of repeated occurrences of a value within a given time range within each pixel. The equation was as follows:

$$P_{NF} = \sum\nolimits_{i=1}^{n} P_{(NF)} = \text{Sum} \left( P1_{(NF)} + P2_{(NF)} + P3_{(NF)} + \ldots + Pn_{(NF)} \right) \tag{3}$$

where $P_{NF}$ is the total number of occurrences of the NDVI value in each pixel, and n is the number of images. Python was used to calculate the pixel-based number of occurrences of each NDVI value from 1 January 2010 to 31 December 2015.

Inundated water persistence time was defined as the number of days in which water was continuously present within a given time range in each pixel. This parameter can be used to quantitatively describe the spatial distribution of hydrological status in a floodplain area, it was calculated using the following equation:

$$P_{WaPe} = \sum\nolimits_{i=0}^{n} P_{(WaPe)} = \text{Sum} \left( P1_{(WaPe)} + P2_{(WaPe)} + P3_{(WaPe)} + \ldots + Pn_{(WaPe)} \right) \tag{4}$$

where $P_{WaPe}$ is the overall water inundation periods of each pixel in the images, unit is day, and n is the number of inundated water days. Daily NDVI images from 1 November 2014 to 31 March 2015 and extracted daily pixel-based water areas were used to acquire a pixel-based floodplain water persistence time map.

## 3. Results

### 3.1. Floodplain Hydro-Geomorphological Dynamics

Water-persistence time and water frequency in the study area from 2010 to 2015 were analysed using the MODIS-NDVI time series. Water persistence time showed an obvious zonal pattern, gradually decreasing from the riverbed to the floodplain margins (Figure 3A). The persistence time of water inundation was 0 days in the extremely rare inundation belt. Among the areas, inundated areas of the riverbed had the longest persistence times, ranging from 108 to 151 days. The belt between the highest persistence time of inundated areas and un-inundated areas had a persistence time ranging from 1 to 97 days. The pixels corresponding to sites that were frequently inundated by floodwaters were used to classify areas as dynamic floodplain areas, and pixels corresponding to sites that were rarely inundated by river water in the study period were used to classify areas as static floodplain areas (Figure 3B).

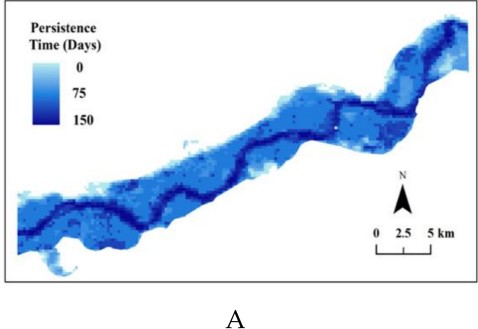
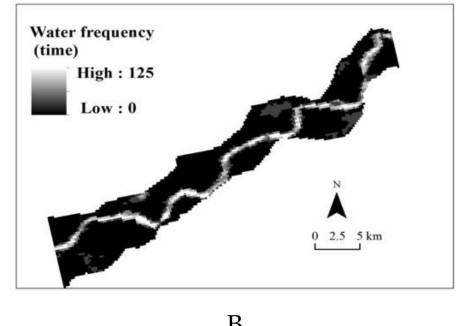

A　　　　　　　　　　　　　　　　　　　　　　　　B

**Figure 3.** Spatiotemporal distribution patterns of water persistence time and frequency. (**A**) Map of floodplain water inundation persistence time map in the study area from 1 November 2014 to 31 March 2015. (**B**) Pixel-based water frequency images in the study area from 2010 to 2015 (from 1 April to 1 November in each year).

The different zonal distributions of water persistence time and water frequency are mainly related to the seasonal variation of water level in the study area. The seasonal patterns of water levels in the study area from 2010 to 2015 were analysed using MODIS-NDVI time-series data. The average water level each year differed significantly between summer and winter during the study period (Figure 4A), being approximately 1050.69 m and 1051.61 m in summer and winter, respectively, approximately 0.91 m higher in winter than in summer. There were also obvious differences in the annual average maximum water level between summer and winter. In the study period, the summer maximum water level was 1051.5 m and the winter maximum water level was approximately 1053 m. The highest yearly maximum summer water level occurred in 2013 and was 1052.17 m (Figure 4B). The monthly average, maximum and minimum water levels between 2010 and 2015 were analysed in the study area. The summer water levels slightly increased from July to October were higher from June to October than that of November and April (Figure 4C).

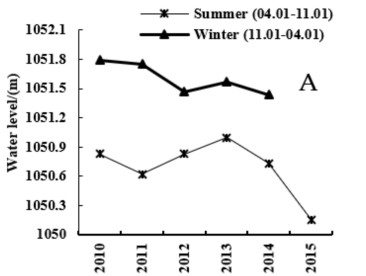
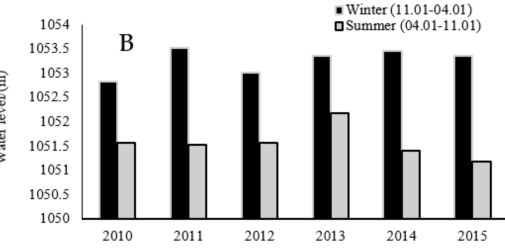
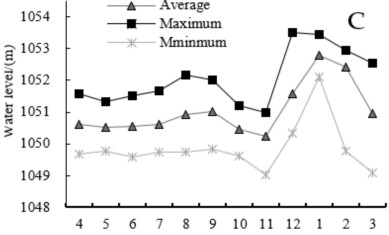

**Figure 4.** The intra- and inter-annual hydrological regime changes at the Bayanganle water stations. (**A**) Yearly average winter and summer water levels at the Bayangaole water station from 2010 to 2015. Winter was defined as November 1 to April 1 of the following year each year; summer was defined as April 1 to November 1 each study year. (**B**) Yearly maximum winter and summer water level at the Bayangaole water station from 2010 to 2015. Winter was defined as November 1 to April 1 of the following year each year; summer was defined as April 1 to November 1 each study year. (**C**) Monthly average, maximum and minimum water levels at the Bayangaole water station from 2010 to 2015.

Based on NDVI values less than 0, the water persistence time was quantified and used to identify the relationships between hydrology and vegetation growth in each pixel of the floodplain. MODIS images were used to extract the water (including ice and snow) area from 1 November 2014 to 31 March 2015. Correspondence analysis was then performed on the daily water levels and the daily cumulative water-inundation area of the study area. The cumulative inundation area changes over time and was significantly correlated with

the daily water level changes (Figure 5). Figure 5 shows that daily average water levels began to rise on 31 November 2014 and that two days later, the inundation area peaked.

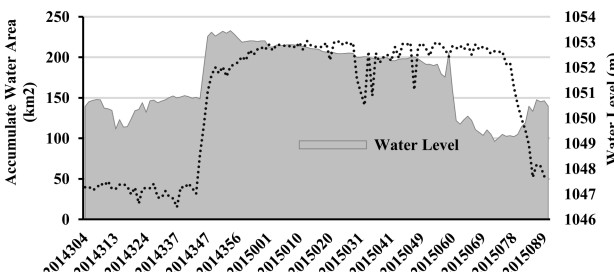

**Figure 5.** Cumulative water inundation area and daily water levels in the study area from 1 November 2014 to 1 April 2015.

*3.2. Dynamics of Floodplain Vegetation*

Pixel-based time-series NDVI from 1 June 2010 to 31 December 2015 were used to illustrate the vegetation response to floodplain hydrogeomorphology, floodplain hydro-geomorphology was significantly stable during this period. Therefore, the NDVI value in each belt was determined based on the zonal distribution differences of floodplain hydro-geomorphological characteristics (Figure 6). Belt A represents the average NDVI value of each pixel located in mostly inundated water and near the average centreline of the river. This belt was almost continuously submerged by water. The NDVI value for this belt was highly dynamic during the entire study period and did not show pronounced interannual changes (Figure 6A). Belt B is the average NDVI value of frequently inundated floodplain belt, located mainly in the riverbed. This belt had very dynamic river hydrological and morphological characteristics, and an NDVI trend shows similar to that of zone A; thus, belt B was irregular and did not exhibit obvious interannual changes. The main vegetation type was grass (Figure 6B). Belt C is the regularly inundated floodplain area and was located at a significant distance from the river. Although it was less affected by summertime hydrological changes than belts A and B, it was regularly inundated by spring and autumn overbanked water caused by ice floods. Belt C had higher NDVI values than the other belts. Figure 6 shows the interannual changes in NDVI. In the regularly inundated belt, the average NDVI value began to gradually increase in April and reached its highest values between July and October; moreover, differences from the other belts are observed. Furthermore, the NDVI values are less than 0 from December to March each year (Figure 6C). Belt D was located in the marginal belt of the floodplain, at the farthest distance from the riverbed (2.3 km). This belt has not been submerged by flooding since the large flooding (5800 m3/s) event that occurred on 24 August 1989. The NDVI curve of belt D presented seasonal dynamics, with the NDVI values increasing from June to September The peak NDVI value of belt D was not as high as that of belt C, and values less than 0 were not observed throughout the study period (Figure 6D).

The NDVI values were plotted for the dynamic and static floodplain areas (Figure 7A), with the blue line representing the static area, and the red line representing the dynamic area. The NDVI values of the static area showed seasonally dynamic changes, whereas those of the dynamic area exhibited a less clear pattern over time. Thus, the static area presented higher peak NDVI values than the dynamic area (Figure 7B) at 0.4 and greater than 0.6, respectively. The range of variation in peak NDVI values was lower for the dynamic area (less than 0.2) than for the static area (between 0.1 and 0.7).

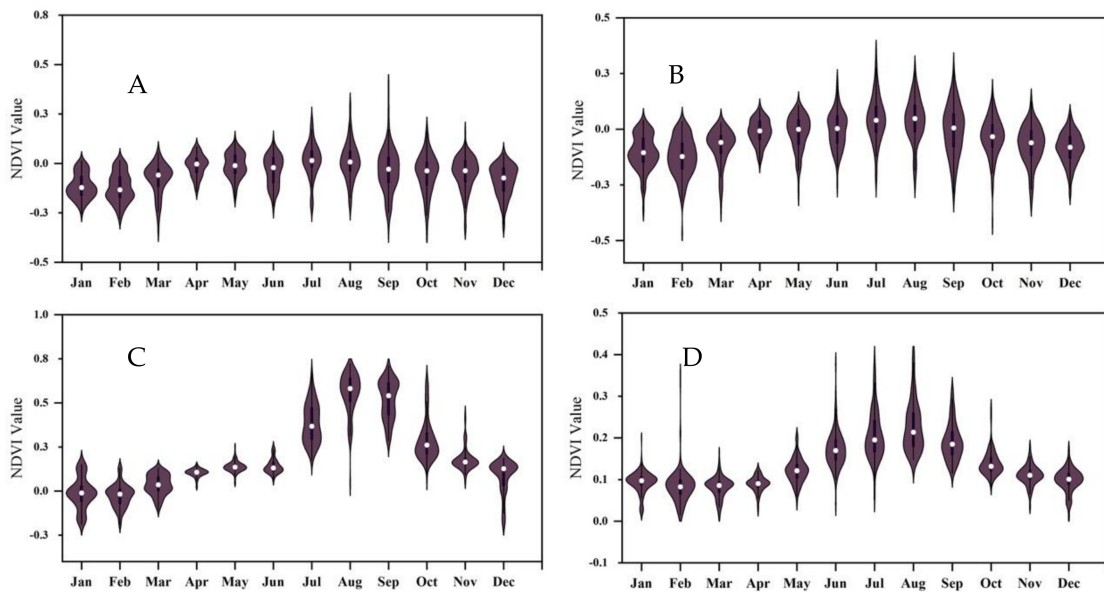

**Figure 6.** Inter-annual changes in the NDVI curves of different zones from the riverbed to the edge of the floodplain. (**A**) represents the continuously inundated floodplain area, (**B**) is the frequently inundated floodplain area, (**C**) represents the regularly inundated floodplain area and (**D**) represents the extremely rare inundated floodplain area.

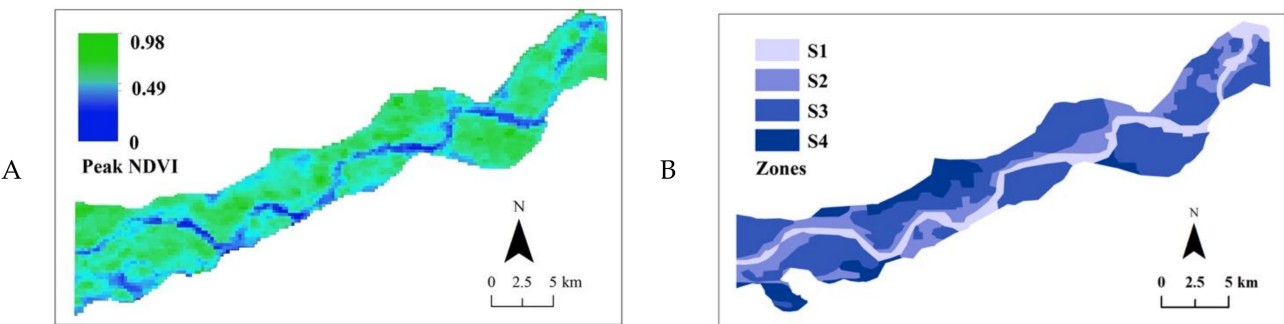

**Figure 7.** Spatiotemporal geomorphological dynamics of NDVI in the study area. (**A**) NDVI over time in different geomorphology units. The red line represents the static geomorphological area, and the blue line represents is the highly dynamic geomorphological area. (**B**) Box chart of the NDVI values in the highly dynamic and static geomorphological areas.

The maximum value calculation model in the ENVI software was used to acquire pixel-based peak NDVI value images from daily multi-band NDVI images from 1 April 2015 to 31 October 2015. The entire study area was divided into four belts based on the different spatial distribution patterns of peak NDVI values. The peak NDVI values, which were less than 0.2, were distributed mainly in the riverbed area in regions S1 and S2, whereas the highest peak NDVI values, which were greater than 0.4, occurred in the S3 region. Peak NDVI values of the marginal plain belts in region S4 were between 0.3 and 0.4, and this region presented a clear border with the belt that containing the highest peak NDVI values throughout the study area. (Figure 8A,B).

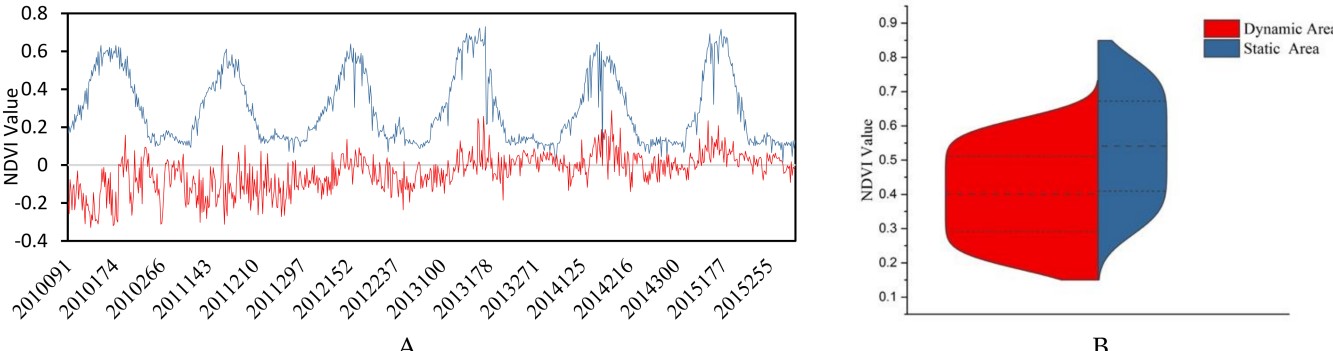

**Figure 8.** Spatial distribution patterns of the peak NDVI values in the study area (**A**) and categories of different zones (**B**), S1 represents the continuously inundated floodplain area, S2 is the frequently inundated floodplain area, S3 represents the regularly inundated floodplain area, and S4 represents the extremely rare inundated floodplain area.

Python was used to calculate the pixel-based maximum possible frequencies of NDVI values from 1 June 2010 to 31 December 2015 in the following bins: >0, 0–0.09, 0.1–0.19, 0.2–0.29, 0.3–0.39, 0.4–0.49, 0.5–0.59 and <0.6. In the map, the pixels coloured by dark blue indicates that the high occurring times of NDVI value in a given range. The colour of pixels from blue to green indicates that the maximum possible frequency of NDVI value in a specified range decreases gradually. A map of the spatial distributions of the maximum possible frequencies of NDVI values less than 0 was produced. NDVI values less than 0 were most frequent in the riverbed belt (Figure 9A). NDVI values between 0 and 0.1 were most frequent in the riverbed, with some appearing in the floodplain margins (Figure 9B). A map of the spatial distribution of the maximum possible frequency of NDVI values between 0.2 and 0.29 was produced. Values in this range occurred in all belts except the riverbed belt, with the highest frequencies distributed mainly in the floodplain margins. The spatial distribution of values in this range was characterised by gradually increasing frequencies from the riverbed to the floodplain margins (Figure 9C). The maps of the spatial distribution of the maximum possible frequencies of NDVI values with ranges between 0.3 and 0.39 and between 0.4 and 0.49 revealed high frequencies mostly in the marginal areas of the floodplain (Figure 9D–F). The high NDVI values, i.e., in the ranges from 0.5 to 0.59 and greater than 0.6, were mainly distributed in the belt between the riverbed and the floodplain margins (Figure 9G,H).

The maximum possible frequency of NDVI values in different ranges was calculated for four belts: belt1, belt 2, belt 3, and belt 4 (Table 2). The continuously inundated floodplain area had a total frequency of NDVI values in average per pixel of 1490 times. The maximum possible frequency of NDVI values less than 0.1 was 1448, representing 97.1% of the total. The maximum possible frequency of NDVI values less than 0 was higher than those of other NDVI ranges in belt 1, with a frequency of 1071. In the frequently inundated floodplain area, the most frequent ranges of NDVI values were <0.1–0.19 and <0.2–0.29, with frequencies twice those in the continuously inundated zone. A value between 0.3 and 0.39 occurred once. In the regularly inundated floodplain area, the range of NDVI values with the highest frequency was the range between 0.1 and 0.19. The maximum possible frequency of NDVI values greater than 0.3 was higher in this belt than in the other three belts. NDVI values greater than 0.4 did not occur in the other three belts, although NDVI values less than 0 was observed in this zone, mainly because of the ice jam phenomenon that occurs every winter. The regularly inundated zone is covered by water or ice in the winter. The extremely rarely inundated floodplain area is the farthest belt from the river. The NDVI values less than 0 only occurred 3 times, and the NDVI ranges from 0 to 0.3 encompassed 95% of the total values.

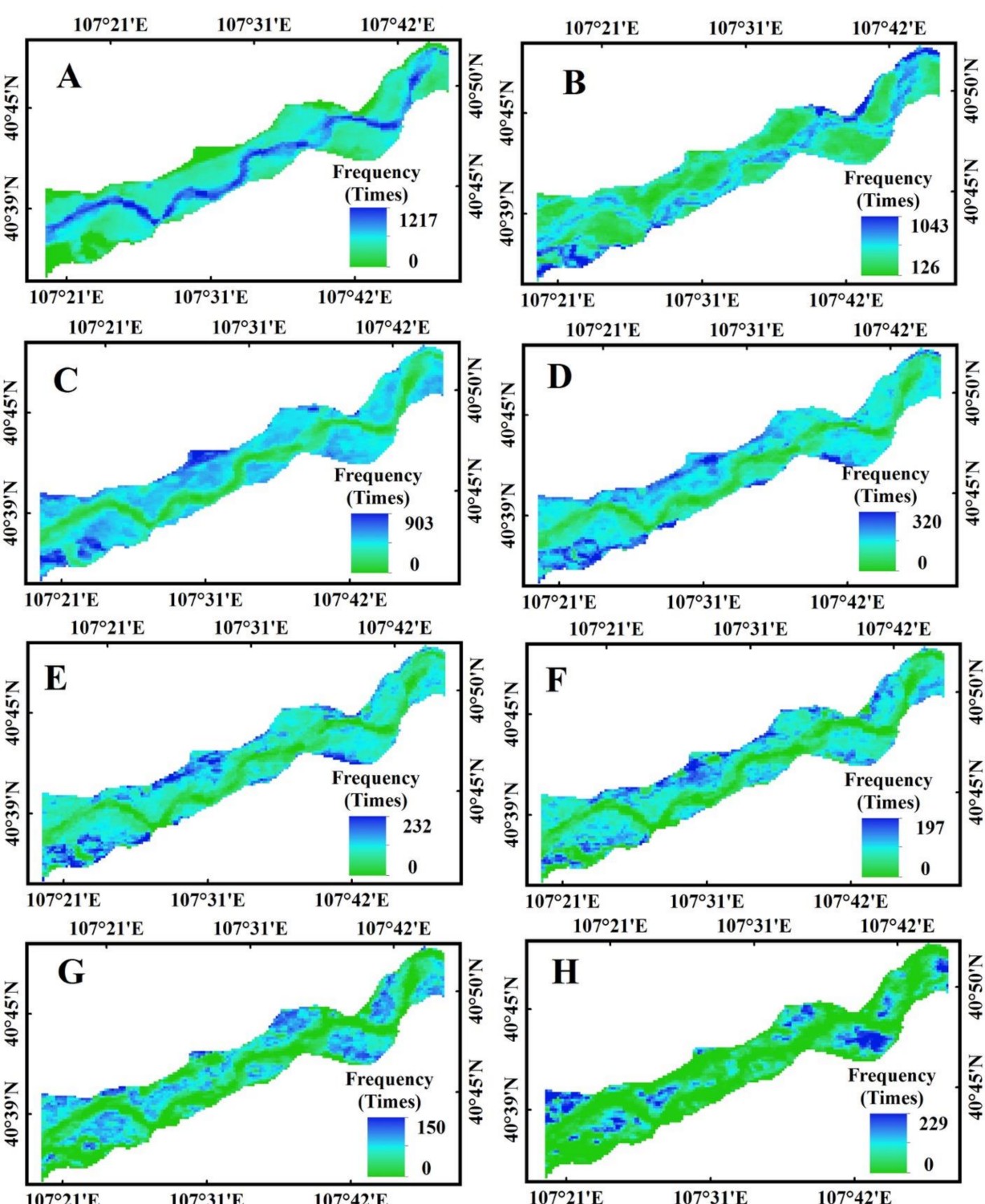

**Figure 9.** Pixel based spatial distributions patterns of the NDVI frequency. (**A**) Frequencies of value < 0, (**B**) 0–0.09, (**C**) 0.1–0.19, (**D**) <0.2–0.29, (**E**) <0.3–0.39, (**F**) <0.4–0.49, (**G**) <0.5–0.59 and (**H**) >0.6.

**Table 2.** The frequency of values in each NDVI ranges in different zones from the riverbed to the edge of three floodplains.

| Belts | NDVI Ranges | | | | | | | | |
|---|---|---|---|---|---|---|---|---|---|
| | <0 | 0–0.09 | 0.1–0.19 | 0.2–0.29 | 0.3–0.39 | 0.4–0.49 | 0.5–0.59 | >0.6 | >0 |
| Continuously inundated area | 1071 | 377 | 34 | 6 | 2 | 0 | 0 | 0 | 419 |
| Frequently inundated area | 1011 | 390 | 79 | 11 | 1 | 0 | 0 | 0 | 481 |
| Regularly inundated area | 330 | 216 | 512 | 143 | 70 | 74 | 88 | 59 | 1162 |
| Extremely rare inundated area | 3 | 794 | 547 | 127 | 21 | 0 | 0 | 0 | 1489 |

The spatial distribution of persistence times of the inundated water area was significantly related to the spatial distributions of NDVI frequencies and peak NDVI values. The spatial distribution of persistence times of inundated water areas between 108 and 151 corresponded to the region of peak NDVI values less than 0.1, and a high frequency of NDVI values less than 0.1 was observed in this region. The belt had a persistence time of water-inundation areas of between 1 and 100 days. Most of the peak NDVI values of this belt were larger than 0.4, and the ranges of NDVI values with the highest frequencies were those encompassing values greater than 0.3. The area with 0 water persistence time in the submerged area was located at the edge of the floodplain; its NDVI peak value was between 0.2 and 0.29, and the NDVI values with the highest frequency were between 0.1 and 0.3 Pixel-based peak NDVI values of the four zones from 2010 to 2015 (from 1 April to 31 October) and water persistence times from 2010 to 2015 (from 1 November to 31 March) were used to establish a simple linear regression model of yearly average peak NDVI values and the yearly winter high-water persistence times (Figure 10). A high correlation coefficient (0.84) was observed between the peak NDVI values of points located in the regular inundation belt and water persistence times between 1 and 100 days. The peak NDVI values of the other belts were not strongly correlated with the persistence time of winter water levels

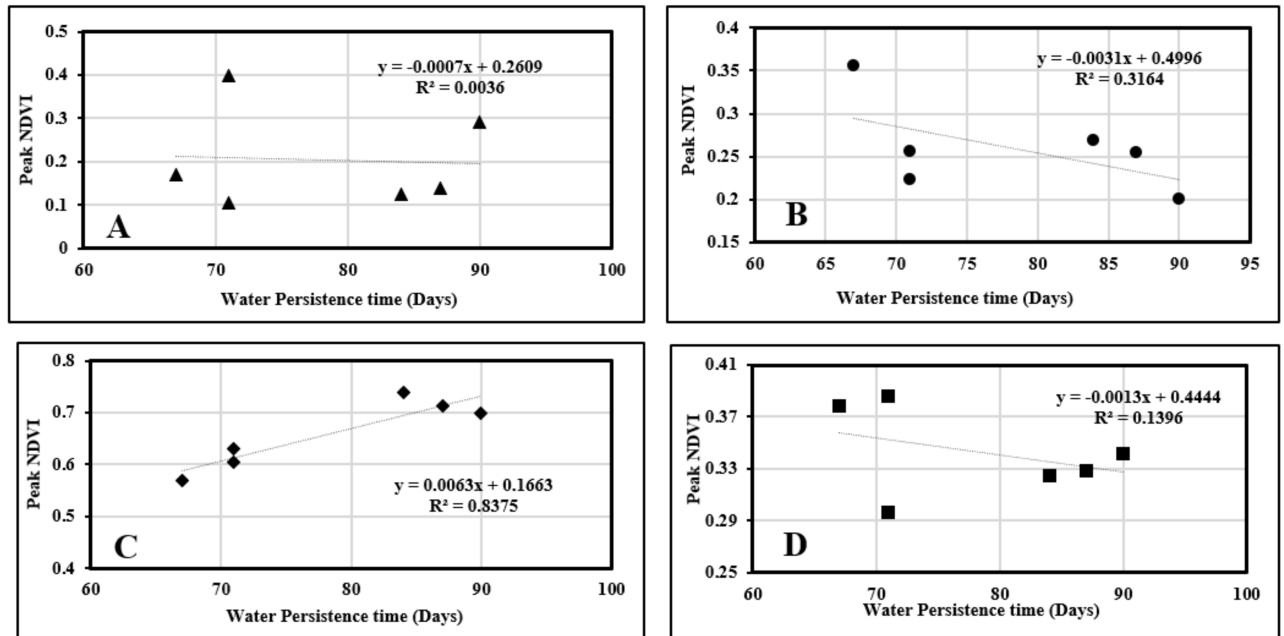

**Figure 10.** Regression analyses of the daily water level and peak NDVI value in different areas. (**A**) for the highly active continuously inundated floodplain belt, (**B**) for the highly active frequently inundated floodplain belt, (**C**) for the static regularly inundated floodplain belt and (**D**) for the static extremely rarely inundated floodplain belt.

## 4. Discussion

### 4.1. Vegetation Responses to Hydro-Geomorphology

The responses of floodplain vegetation to hydro-geomorphology are of significance for investigating the eco-hydrology of arid-region floodplains [6,52,53]. Here, we divided the study area, located in the wandering reaches of the Yellow River, into four distinct hydro-geomorphological belts. We found that the vegetation had varying responses among the different hydro-geomorphological belts from the river one bend to the floodplain interior. For example, the average peak NDVI value of the regular water inundation belt was 0.4, significantly higher than the corresponding values of the frequently and extremely rare water inundation belts, which is inconsistent with Marchetti et al. (2016), who found that NDVI patterns have distinct characteristics among in the elevation, flood dynamics and geomorphological differences [54]. In their study, they divided the study area in the Paraná River floodplain into two parallel belts based on the NDVI patterns. This discrepancy is probably due to the different flooding dynamics analysed in the two studies, in this study reach, ice-jam flooding occurs each spring and autumn that contributing to the complex hydrological regime of this floodplain region. In this study, there was a significant correlation between the water persistence time and the maximum NDVI value of each pixel in the regular water inundation area ($\rho > 0.8$, Figure 10). This finding suggests that regular water inundation is beneficial to vegetation growth and biomass [17].

### 4.2. Remote-Sensing Application

Remote sensing represents an unparalleled tool for investigating floodplain vegetation and river hydro-geomorphology in arid regions [26,55,56]. For highly dynamic, wandering river floodplains, performing field surveys or historical mapping to capture the short-term dynamics of floodplains is costly, and obtaining highly dynamic and detailed characteristics of large-scale floodplain areas is difficult with these techniques. The use of remote-sensing images with high-revisit frequency can overcome these challenges [29]. Our study was based on daily MODIS9GQ-NDVI images with a 250 m spatial resolution, and a pixel-based analysis of the vegetation dynamics from the river centreline to the edge of the floodplain was performed. Although the vegetation responses to river hydrology and geomorphology using remote-sensing technology have been debated [17,26,56], this technology allowed the quantification of vegetation growth and biomass using NDVI time-series statistical indicators, such as maximum NDVI value and NDVI frequency. Pixel-based differences in water-inundation persistence times in each floodplain hydro-geomorphological belt were quantified to provide investigate the spatial distribution characteristics of floodplain vegetation-hydrogeomorphology from the riverbed to the edge of the floodplain. Mohammadi et al. (2017) used time series of remotely sensed normalized difference water, vegetation and moisture indices based on MODIS images to characterise the floodplain extinction [26]. They found that this type of approach is a complementary tool for ecological studies of floodplain productivity, which have mainly used vegetation indices to study the spatiotemporal effects of flooding on productivity. However, in the present study, the study area included few vegetation canopies or types (being covered mostly by herbs) and was situated in a medium-sized floodplain; thus, differences in spatial distributions would not have been detectable using low spatial-resolution remote-sensing images, such as those with 500 m spatial resolution. Therefore, in areas such as the present study area, MODIS-NDVI time series with a 250 m spatial resolution have greater potential than those of lower resolution for quantifying the changes in vegetation different distribution and growth that occur in response to floodplain hydro-geomorphological dynamics. Our results can help improve the management of the Yellow River floodplains located in arid regions. Ablat et al. (2019) used different Landsat time-series images and hydrological data and proposed that the responses of wetland landscape types in different zones to floodplain hydrology varied under different dam operations [17]. They also identified the limitations of spatial resolution of the MODIS data in their study. In the present investigation, we found that the responses of floodplain vegetation to hydro-geomorphology in the study area

varies among zones of the natural vegetation from the riverbed to the floodplain margins. Due to the high coverage of natural vegetation, the MODIS times series with high spatial resolution were well suited to this study. Although MODIS250 time resolution can meet the demand of this study, if we want to obtain more accurate results, the omission error will become one of the fatal defects. The ideal way to solve this problem is to combine the use of remote sensing images with different time frequencies, such as Sentinel images with high spatial-temporal resolution launched in recent years. The combined use of Sentinel, Landsat and SPOT sensors can provide a remote sensing database with high temporal and spatial resolution, but its large amount of data records is a problem. Moreover, these new data sources cannot provide a powerful historical dataset and have no advantages in detecting environmental changes. Therefore, MODIS images are the most ideal data source to study the relationship between long time series vegetation spatio-temporal changes and river hydrogeomorphology.

## 5. Conclusions

Time-series daily MODIS-NDVI images were used to quantify pixel-based floodplain vegetation responses to changes in floodplain hydrogeomorphology from the riverbed to the edge of the floodplain in arid regions. Using MODIS-NDVI time series statistical indices, including the frequency of NDVI values, peak NDVI value and water persistence time, to quantify vegetation responses to hydrogeomorphology in highly dynamic wandering rivers in arid regions represents a considerable advance. Floodplain vegetation had obvious spatial zonal distribution characteristics from the riverbed to the edge of floodplains, vegetation growth condition in the regularly inundated floodplain belt is better than that in other belts, and the average maximum NDVI value is 0.23, which is significantly higher than that in other belts, indicating that vegetation growth and biomass were mainly controlled by floodplain hydrogeomorphology. There is a significant correlation between the water persistence times and peak NDVI values, the correlation coefficient is 0.84, at the significance level of 0.05 showed that regular water inundation is better than high-frequency inundation or extremely rare inundation for floodplain vegetation growth. Within regular water persistence time scales, increasing the water-inundation time could benefit vegetation growth and biomass in regularly inundated static areas. Overall, our results suggest that the vegetation greenness response to the river hydro-geomorphology is different from the river to the edge of the floodplain. Thus, a better understanding of the interactions between the floodplain vegetation and river hydro-morphology and river water resource management in arid-region floodplains.

**Author Contributions:** Conceptualization, X.A. and G.L.; software, X.A.; validation, X.A., Q.L. and C.H.; formal analysis, X.A.; writing—original draft preparation, X.A.; writing—review and editing, X.A. and C.H.; funding acquisition and project administration, G.L. and C.H. All authors have read and agreed to the published version of the manuscript.

**Funding:** This research was funded by the International Cooperation Projects of the National Natural Science Foundation Committee of China (Impact of Climate and Land Use Change on Water Quantity and Water Quality in the Mum River Basin under), grant number 41661144030 and the Special Project of Lancang-Mekong River Cooperation of the Ministry of Science and Technology of the People's Republic of China.

**Institutional Review Board Statement:** Not applicable.

**Informed Consent Statement:** Not applicable.

**Data Availability Statement:** Not applicable.

**Acknowledgments:** We thank LAADS DAAS and AIR for the remote sensing data. Special thanks for the helpful comments and suggestions from two anonymous reviewers. Thanks again for the time and effort you put into our paper.

**Conflicts of Interest:** The authors declare no conflict of interest.

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
