# Peer review of "Using MODIS-NDVI Time Series to Quantify the Vegetation Responses to River Hydro-Geomorphology in the Wandering River Floodplain in an Arid Region"

_water, doi:10.3390/w13162269_

Round 1
Reviewer 1 Report
Arid and semi-arid areas constitute over 30% of the world’s land surface, in these water-limited ecosystems, soil moisture strongly affects land surface hydrology, subsurface hydrology, and eco-hydrological fluxes. This study will be useful in such kind of region. However, there is a need to improve the manuscript before considering it for publication. In the entire manuscript, there are few points which authors need to consider as it is important in studying the feedback between vegetation and soil moisture especially in semiarid and arid areas which are water limited ecosystems. Though authors have identified the research gaps the literature survey part can be more streamlined and while coming towards the problem statement. The introduction section needs some rework and restructuring to make a precise outline of the study
Currently, some of the statements are not supported by published works. Authors may like to find studies in line with their statements to add scientific weight to their observations. I believe that after duly addressing the comments authors can improve the quality of the manuscript substantially to make it more insightful.
Line 31-32 Authors have not have mentioned how there exists the interdependence of topographic features and hydrology which in turn affects the floodplain. I would recommend authors add a sentence in the Introduction “Differences in landscape shapes significantly rely on the amount of water and the geomorphic processes that affect the runoff in the basin” (Srivastava et al., 2021)
Srivastava, A., Saco, P. M., Rodriguez, J. F., Kumari, N., Chun, K. P., & Yetemen, O. (2021). The role of landscape morphology on soil moisture variability in semi‐arid ecosystems. Hydrological Processes, 35(1), e13990.
Line 37-38 There have been various studies which have studied the influence of fluvial and diffusive processes on vegetation (Srivastava et al., 2021). Just one or two reference gives the impression that the extensive literature review is not being conducted and it is needed to address the problem. Hence, I would strongly recommend adding these recent and important references to add more scientific weight in their Introduction in the lines mentioned above.
Line 48-69 Authors have not discussed about the selection of NDVI index over other indices such as SAVI, EVI etc. It would be better to provide the advantages and disadvantages of using this index. I have a big concern of using NDVI has several drawbacks such as topographic illumination, shading effect, solar angle issues which has not been discussed and elaborated. These aspects of NDVI has been mentioned and explained clearly in one of the recent paper by Kumari et al., 2020 from which authors can benefit.
Kumari, N., Saco, P. M., Rodriguez, J. F., Johnstone, S. A., Srivastava, A., Chun, K. P., & Yetemen, O. (2020). The grass is not always greener on the other side: Seasonal reversal of vegetation greenness in aspect‐driven semiarid ecosystems. Geophysical Research Letters, 47(15), e2020GL088918.
Authors have missed to include the drawbacks of other satellite products such as AVHRR, LANDSAT based indirect vegetation proxies. I encourage and recommend the authors to incorporate a comprehensive detailed review related to the use of satellite-based remote sensing products in the estimation of vegetation growth. Authors may like to find studies in line of their statements to add the scientific weight in their observations. There is a vast literature on this topic.
Please provide the latitude and longitude for the Figure 1
Authors have no where mentioned the LANDSAT in the introduction if they have utilised, it should be mentioned in the Introduction. And why it is not mentioned in the title of the paper.
Author Response
Reviewer #1:
We are very grateful to your comments for the manuscript. According with your advice, we amended the relevant part in manuscript. All of your questions were answered below.
- Line 31-32 Authors have not have mentioned how there exists the interdependence of topographic features and hydrology which in turn affects the floodplain. I would recommend authors add a sentence in the Introduction “Differences in landscape shapes significantly rely on the amount of water and the geomorphic processes that affect the runoff in the basin” (Srivastava et al., 2021)
Response: Thank you for your recommendation, we have added a sentence you recommended in the introduction and added the reference accordingly(on line 30-33, page 1 in modified revision)
- Line 37-38 There have been various studies which have studied the influence of fluvial and diffusive processes on vegetation (Srivastava et al., 2021). Just one or two reference gives the impression that the extensive literature review is not being conducted and it is needed to address the problem. Hence, I would strongly recommend adding these recent and important references to add more scientific weight in their Introduction in the lines mentioned above.
Response: Special thanks for reviewer's patient advice, we are very grateful. Relevant literatures that recently published has been added. The literatures were added as follows:
[1] Ak, A. , Hm, B. , & Zbcd, E. (2020). Spring vegetation green-up dynamics in central europe based on 20-year long modis ndvi data. Agricultural and Forest Meteorology, 287. https://doi.org/10.1016/j.agrformet.2020.107969
[2] Am, A. , Jpj, A. , & Mwd, B. (2020). Riparian vegetation as an indicator of stream channel presence and connectivity in arid environments - sciencedirect. Journal of Arid Environments, 178. https://doi.org/10.1016/j.jaridenv.2020.104167
[3] Diehl, R. M. , Wilcox, A. C. , & Stella, J. C. . (2020). Evaluation of the integrated riparian ecosystem response to future flow regimes on semiarid rivers in colorado, usa. Journal of Environmental Management, 271(319), 111037.
https://doi.org/10.1016/j.jenvman.2020.111037
[4] Diehl, R. M. , Wilcox, A. C. , & Stella, J. C. . (2020). Evaluation of the integrated riparian ecosystem response to future flow regimes on semiarid rivers in colorado, usa. Journal of Environmental Management, 271(319), 111037.
https://doi.org/10.1016/j.jenvman.2020.111037
[5] Ding, J., Johnson, E.A., Martin, Y.E., 2018. Linking Soil Moisture Variation and Abundance of Plants to Geomorphic Processes: A Generalized Model for Erosion-Uplifting Landscapes. J. Geophys. Res. Biogeosciences 123, 960–975. https://doi.org/10.1002/2017JG004244
[6] Dari, J., Morbidelli, R., Saltalippi, C., Massari, C., Brocca, L., 2019. Spatial-temporal variability of soil moisture: Addressing the monitoring at the catchment scale. J. Hydrol. 570, 436–444. https://doi.org/10.1016/j.jhydrol.2019.01.014
[7] Baartman, J.E.M., Temme, A.J.A.M., Saco, P.M., 2018. The effect of landform variation on vegetation patterning and related sediment dynamics. Earth Surf. Process. Landforms 43, 2121– 2135. https://doi.org/10.1002/esp.4377
[8] Kumari, N., Saco, P. M., Rodriguez, J. F., Johnstone, S. A., Srivastava, A., Chun, K. P., Yetemen, O., 2020. The Grass Is Not Always Greener on the Other Side: Seasonal Reversal of Vegetation Greenness in Aspect‐D riven Semiarid Ecosystems. Geophys. Res. Lett., 47(15), e2020GL088918. https://doi.org/10.1029/2020GL088918
[9] Nagler, P. L. , A Barreto‐Muoz, Borujeni, S. C. , Jarchow, C. J. , MM Gómez‐Sapiens, & Nouri, H. , et al. (2020). Ecohydrological responses to surface flow across borders: two decades of changes in vegetation greenness and water use in the riparian corridor of the colorado river delta. Hydrological Processes, 34.
https://doi.org/10.1002/hyp.13911
[10] White, H. A. , Scott, J. K. , & Didham, R. K. . (2021). Evidence of range shifts in riparian plant assemblages in response to multidecadal streamflow declines. Frontiers in Ecology and Evolution, 9.
https://doi.org/10.3389/fevo.2021.605951
[11] Wulder, M. A. , Loveland, T. R. , Roy, D. P. , Crawford, C. J. , Masek, J. G. , & Woodcock, C. E. , et al. (2019). Current status of Landsat program, science, and applications. Remote Sensing of Environment, 225, 127-147.
https://doi.org/ 10.1016/j.rse.2019.02.015
[12] Xiong, X. , Angal, A. , Chang, T. , Chiang, K. , & Wu, A. . (2020). MODIS and VIIRS calibration and characterization in support of producing long-term high-quality data products. Remote Sensing, 12(19), 3167.
https://doi.org/10.3390/rs12193167
[13] Xiong, X. , & Butler, J. J. . (2020). MODIS and VIIRS calibration history and future outlook. Remote Sensing, 12(16), 2523.
https://doi.org/10.3390/rs12162523
[14] Luo, D., Mao, W., Sun, H., 2017. Risk assessment and analysis of ice disaster in Ning-Meng reach of Yellow River based on a two-phased intelligent model under grey information environment. Nat. Hazards 88 (1), 591–610.
https://doi.org/10.1007/s11069- 017-2883-6.
- Line 48-69 Authors have not discussed about the selection of NDVI index over other indices such as SAVI, EVI etc. It would be better to provide the advantages and disadvantages of using this index. I have a big concern of using NDVI has several drawbacks such as topographic illumination, shading effect, solar angle issues which has not been discussed and elaborated. These aspects of NDVI has been mentioned and explained clearly in one of the recent paper by Kumari et al., 2020 from which authors can benefit.
Response: We agree with this suggestion and explain in detail the application goal of NDVI index in this study. Based on the description of this term in section one (Introduction), lines 60-65 of the original paper, we made it clearer as follows(on line 60, page 2 in modified revision):
NDVI is the most commonly used vegetation index, which compares the reflectance intensity of red band and near-infrared band to quantify green living vegetation(Rouse et al., 1973). Through the band intensity ratio, NDVI removes a large amount of noise caused by cloud shadow, terrain and solar angle changes, as well as atmospheric attenuation in visible and infrared bands, which makes NDVI less sensitive to lighting conditions (Huete et al., 2002; Justice et al., 1981; Martín‐Ortega et al., 2020).
- Authors have missed to include the drawbacks of other satellite products such as AVHRR, LANDSAT based indirect vegetation proxies. I encourage and recommend the authors to incorporate a comprehensive detailed review related to the use of satellite-based remote sensing products in the estimation of vegetation growth. Authors may like to find studies in line of their statements to add the scientific weight in their observations. There is a vast literature on this topic.
Response: We would like to thank the referees for their helpful comments. The advantages and disadvantages of other sensors in extraction of the vegetation greenness are described. Related information is contained in our revised manuscript. (lines 50-58, page 2):
Readily available Landsat imagery with a long record provides sufficient spatial resolution to capture subtle dynamics of vegetation at the landscape scale but its coarse time resolution and interference of clouds makes acquisition of a detailed time-series of vegetation greenness over a short time period unlikely(Haas et al., 2009). The two sentinel 2 satellites that launched in recent years, like Landsat, can be obtained free of charge all over the world, and also has a high spatial resolution (10-60m). Although the spatial resolution is highly consistent with Landsat, but the revisit time is reduced to 5 days. MODIS(Moderate Resolution Imaging Spectroradiometer) images which 1-day frequency can reflect the change of vegetation greenness in highly dynamic river channel, but their coarse spatial resolution provides a sub-optimal representation of the instantaneous distribution of vegetation greenness, especially in flat landscapes with complex channel networks (Aires et al., 2014; Huang et al., 2013; Justice et al., 1998; Ogilvie et al., 2015; Sakamoto et al., 2007).
- Please provide the latitude and longitude for the Figure 1
Response: We are sorry for this negligence. We added the latitude and longitude to the Figure 1.
- Authors have no where mentioned the LANDSAT in the introduction if they have utilised, it should be mentioned in the Introduction. And why it is not mentioned in the title of the paper.
Response: Thanks for the Reviewer’s thoughtful comment. The content related to Landsat data has been added to the introduction. In this study, Landsat data are only used for the division of different floodplain zones, and have not been applied to the quantitative analysis of the relationship between vegetation greenness and river hydro- geomorphology. Therefore, Landsat data are not mentioned in the title of this paper.

Reviewer 2 Report
Dear Authors,
the manuscript is rather interesting and well-written, even if the used methods are not new and the overall novelty is reduced.
Here some general comments, while more detailed notes can be found in the attached pdf.
Introduction
Please expand it, considering more up-to-date references and pointing out why your study is timely. In the present version, the novelty is not clear enough.
Sections 2 and 3
I suggest combining these two sections, entitling them "Materials and Methods" (a more common approach).
In general, there is a need for more details, to allow for the study reproducibility. Similarly, please improve the presentation of the study rationale (e.g., why you choose only 3 images, and 2 of them from the same years, etc.).
Please provide a few more info about the river hydrology (e.g., hydrograph).
Results
This section is rather ok, but there is the need for having more clear figures.
Discussion
Besides what you already discussed, please provide some comments on the quality of the data, as well as on the possible steps needed for obtaining more precise estimates of the river/vegetation dynamics (e.g., deriving NDVI from the Sentinel dataset).
Conclusions
Please add a summary of the results, comparing them with the initial motivation of the study. Have you answered all the research questions described in the Introduction? If yes, how? If not, what are the next steps needed for answering them?

Author Response
Reviewer #2:
Thanks for your good comments on our paper. We have revised our paper according to your comments:
- Please expand it, considering more up-to-date references and pointing out why your study is timely. In the present version, the novelty is not clear enough.
Response: Thanks again. This comment was a big help in improving the introduction part of the paper. The latest research works have been added in the introduction section and highlight the innovation of the paper. The newly added recent relevant references have been listed in the answer to the second question from the Reviewer 1.
- Study area and method. I suggest combining these two sections, entitling them "Materials and Methods" (a more common approach).
Response: Thank you for your advice, we combined two sections that you mentioned.
- In general, there is a need for more details, to allow for the study reproducibility. Similarly, please improve the presentation of the study rationale (e.g., why you choose only 3 images, and 2 of them from the same years, etc.).
Response: Thanks for the Reviewer’s thoughtful comment. The reasons why we choose these three periods images were given in the section 2.3 and were highlighted in green color.
Related information is contained in our revised manuscript. (lines 47-49, page 5):
the intra-annual hydrological periods of high flow, low flow, extremely low flows and ice-jam flooding were determined from middle-resolution Landsat images and the hydrological data.
As for why only three images are selected, and of which two are from the same year, the main reason is that according to the annual change of river flow, we used these three images to divide the river channel into four different belts. The image from 2015 were applied to extract the ice cover area in the study area. This study refers to the maximum ice coverage area in 2014,due to the poor image quality in the winter of 2014, the ice coverage in March 2015 was replaced by the ice cover area in 2014.
- Please provide a few more info about the river hydrology (e.g., hydrograph).
Response: Thanks again. The information about the river hydrology has been added. Related information is contained in our revised manuscript. (lines 119-120, page 3):
During the study period(2010-2015), the interannual hydrological regime has been typically stable. Summer water levels slightly increased between July and Oct each year. Because of the geographical conditions, the channel depth has decreased, and flows from low to high latitudes together have caused ice flooding phenomena each year (Luo et al., 2017). In winter, the downstream areas freeze earlier than does the upstream portion of the study reach, which causes the water level to rapidly increase, Thus, the river water in the study reach experiences a “banking” phenomenon each winter (Tang et al., 2016). The floodplain area is inundated with water during this period, and low-temperature conditions cause inundated waters to quickly freeze in floodplain areas rather than re-entering river channels.
- This section is rather ok, but there is the need for having more clear figures.
Response: It is very true, as you suggested, that the quality of most of the figures in this paper is very poor. We improved the quality of all of the figures mentioned by you and clear figures were provided separately as supplementary documents. We are grateful to you for pointing out these deficiencies in our paper.
- Besides what you already discussed, please provide some comments on the quality of the data, as well as on the possible steps needed for obtaining more precise estimates of the river/vegetation dynamics (e.g., deriving NDVI from the Sentinel dataset).
Response: Thanks for your kind advice. We carefully considered this point when we answered each comment, and the corresponding changes in the manuscript were added after the reviewer comment along with locations in the manuscript (line 431-439 and page16):
Although MODIS250 time resolution can meet the demand of this study, but if we want to obtain more accurate results, the omission error will become one of the fatal defects. The ideal way to solve this problem is to combined use of remote sensing images with different time and spatial frequencies, such as Sentinel images with high spatial-temporal resolution launched in recent years with high time resolution. The combine use of Sentinel, Landsat and SPOT sensors can provide remote sensing database with high temporal and spatial resolution, but its large amount of data records is a problem. Moreover, these new data sources cannot provide a powerful historical dataset, have no advantages in detecting environmental changes. therefore, MODIS images is the most ideal data source to study the relationship between long-time series vegetation spatio-temporal changes and river hydrogeomorphology.
- Please add a summary of the results, comparing them with the initial motivation of the study. Have you answered all the research questions described in the Introduction? If yes, how? If not, what are the next steps needed for answering them?
Response: Thanks for the Reviewer’s thoughtful comment. Your comment has greatly helped to improve the paper quality.
The detailed revision can be found on lines 447-453, Page 16:
Floodplain vegetation had obvious spatial zonal distribution characteristics from the riverbed to the edge of floodplains, vegetation growth condition in the regularly inundated floodplain belt is better than that in other belts, and the average maximum NDVI value is 0.23, which is significantly higher than that in other belts, indicating that vegetation growth and biomass were mainly controlled by floodplain hydrogeomorphology. There is a significant correlation between the water persistence times and peak NDVI values, the correlation coefficient is 0.84, at the significance level of 0.05 showed that regular water inundation is better than high-frequency inundation or extremely rare inundation for floodplain vegetation growth.
For comprehensively answering the questions raised in the introduction, according to the hydro-geomorphological characteristics of the floodplains, we used time series MODIS images, applied typical statistical analysis, GIS spatial statistics, computer technologies to analysis the relationship between the floodplain vegetation greenness and hydro-geomorphological dynamics.

Round 2
Reviewer 1 Report
I want to thank the authors for addressing previous comments and for their constructive work. The authors have addressed all previous concerns expressed by the reviewers and in the process have improved the work, confirmed the validity of their findings and gained confidence in their introduction, methods, results and conclusions. I would like to congratulate the authors for an interesting and well executed work and I recommend this manuscript for publication in Water (MDPI) in its current form.
Author Response
Thank you to your time and energy for our paper.
Reviewer 2 Report
Dear Authors,
thank you very much for having (partially) updated the manuscript following my comments.
However, I am wondering if you received also the commented pdf, as your answers refer only to my general comments, while detailed suggestions (in the pdf) were not addressed. If you received it, please revise the text once again, considering all my comments (both general and detailed) in a more thorough manner.
Author Response
List of Responses
Dear Dr. Avy Guo and anonymous reviewer 2:
Thank you very much for your time for our article and the Reviewer's evaluation and comments on our paper entitled “Using MODIS-NDVI Time Series to Quantify the Vegetation Responses to River Hydro-geomorphology in the Wandering River Floodplain in Arid Region” (ID: water-1293040). We have studied the comments carefully and made corrections that we hope will meet with your approval. Revised portions are marked in marked in the paper. In this version, mainly according to the reviewer’s comments marked in the article, the manuscript was revised once again.
The detailed corrections in the paper and the responses to comments from Reviewer2 are as follows:
Reviewer #2:
We are very grateful to your comments for the manuscript. According with your advice, we amended the relevant part in manuscript. All of your questions were answered below.
- Actully, these three elements are paramount in all the ecosystems.
Response: Thank you for your reminder. The words “in dry areas” in this sentence has been changed to “on earth”.
- Please follow the journal guidelines for formatting the references.
Response: Thank you for your advice. All reference formats have been revised according to the requirements of this journal.
- What do you mean? Lines 33-35.
Response: Thanks for the Reviewer’s reminder. This sentence have been changed as follows: “Differences in landscape shapes significantly rely on the amount of water and the geomorphic processes that affect the runoff in the basin” based on the reviewer’s suggestion. Marked on Lines 30-31,Page 1 in the new version of paper.
- I can assume that the study period was 2010-2015,but please state it explicitly for improving the paper readability.
Response: Thanks again. The sentence was changed as follows: The yearly mean precipitation for the study area was 150-400mm from 2010 to 2015, and three quarters of the annual rainfall is concentrated in summer (between June and September). During this period, the mean annual air temperature was 8.2°C, and the regions experience 135-150 frost-free days and 3100-3300h of sunshine per year. Lines 110-113, Page 3.
- Line 96: strong geomorphological dynamics. Question: which ones? Any references?
Response: We refer here to highly dynamic channel morphology. The reference was added corresponding position (Su et al.,2017)
[47]Su, T., Wang, S., Mei, Y., Shao, W., 2015. Comparison of channel geometry changes in inner Mongolian reach of the Yellow River before and after joint operation of large up- stream reservoirs. J. Geogr. Sci. 25 (8), 930–942.
https://doi.org/10.1007/s11442- 015-1211-x.
- Please improve the figure quality .
Response: Thanks for your kind advice. All figures in the article have been updated.
- Which hydrological measurements? Only daily water level as described below? Only one gauge was considered?
Response: Thanks again. Mainly refers to the river runoff and water level. Yes, the Bayangaole water station was considered in this paper.
- Add some details(e.g., spatial resolution)
Response: Thanks for you kind advice. Details conceptions were added in the manuscript and in the Table 1.
- Please follow a chronological order. Moreover, why did you chose only these images? Please provide a rationale.
Response: Thanks for the Reviewer’s thoughtful comment. The reasons why we choose these three periods images were given in the section 2.3 and were highlighted in yellow color.
Related information is contained in our revised manuscript. (lines 47-49, page 5):
the intra-annual hydrological periods of high flow, low flow, extremely low flows and ice-jam flooding were determined from middle-resolution Landsat images and the hydrological data.
- Line 132, measured were? Above you presented only water levels.
Response: This content has been added to the corresponding section of the article.
Related information is contained in our revised manuscript. (Line 160, Page 5):
The average daily water level and run off data used in this study were collected at the Bayangaole water station that located in the upstream region of the study area (40° 19′ N, 107° 02′ S) (Fig.1), and were obtained from the Yellow River Conservancy Commission (YRCC).
- Please add a table with water levels/discharges for the three analyzed Landsat images.
Response: Thanks for your advice. The table was added below the section 2.3 Lateral zonal distributions of the study area of the manuscript.
- Please follow the journal guidelines for writing the equestions.
Response: Thanks again. The format of all equations in the paper were adjusted according to the requirements of the journal.
- Please add more details on the river hydrology. Is 731m3/s the annual mean discharge? What dose IHA mean?
Response: Thanks again. The information about the river hydrology has been added.
Related information is contained in our revised manuscript. (lines 119-120, page 3):
During the study period(2010-2015), the interannual hydrological regime has been typically stable. Summer water levels slightly increased between July and Oct each year. Because of the geographical conditions, the channel depth has decreased, and flows from low to high latitudes together have caused ice flooding phenomena each year (Luo et al., 2017). In winter, the downstream areas freeze earlier than does the upstream portion of the study reach, which causes the water level to rapidly increase, Thus, the river water in the study reach experiences a “banking” phenomenon each winter (Tang et al., 2016). The floodplain area is inundated with water during this period, and low-temperature conditions cause inundated waters to quickly freeze in floodplain areas rather than re-entering river channels.
731m3/s is the boundary division of high flow and low flow in the study area in 2014.
IHA means is the Index of Hydrological Alternation, given in the line 164, page5.
Thanks again for reviewers questions, thank you for pointing out. The relevant contents are more detailed. Thanks again.
- Could you please explain better this concept? Are you assuming that floodplains frequently inundated are more dynamic under a morphological point of view, with respect to relatively arid environments? Can you please provide some references/examples to sustain this assumption?
Response: Thanks again. This expression method is mentioned in the Gurnell et al., 2016. Our basis is that the highly dynamic river flow in arid areas are submerged different flooded areas within the year, and its has interannual regularity. We spatially express these areas by used remote sensing and hydrological situation analysis, as a standard of transect division from river to flood plain edges.
[22]Gurnell, A. M., Grabowski, R. C. (2016). Vegetation–hydrogeomorphology interactions in a low‐energy, human‐impacted river. River Research and Applications, 32(2), 202-215.
https://doi.org/10.1002/rra.2922
- Please avoid shortening the names of months in the main text
Response: Thank you for taking our paper seriously. All the shortening of the names of months have been changed in the main text.
- Why have you selected these classes and not other? Please provide a few more details on the study rationale
Response: Thank you for your thoughtful comments. The value range of NDVI in the entire study area is 0 ~ 0.6. And we divided it into 8 small scales per 0.1 unit. And compared and analyzed the spatial distribution pattern of the cumulative maximum occurrence times of NDVI from the edge of floodplain to the river in different value ranges.
- I suggest keeping the same scale(0-1217) in the figure9. This can help in comparing the different frequencies
Response: Thanks again. In this study, the distribution pattern of the cumulative maximum frequency of different NDVI value from the floodplain edge to the river were compared and analyzed, rather than the distribution of different vegetation indices in the whole study area. Therefore, we concerned that different legends are more comparative in the distribution patterns of the cumulative maximum frequency in different ranges of NDVI values.
- You can try to represent these values in a relative manner(e. frequency/ max possible frequency).
Response: Thanks you for your constructive comments. We also found that it should be better to describe it in relative manner after your comments. According to your suggestion, we used the maximum possible frequency to described it.
- I suggest moving some of these sentence to the Introduction, and the rest to the Conclusions. Usually, the purpose of the study is not reported in the Discussion section, while here you have to show that your research fits the state-of-art, and fills some gaps in our understanding of river dynamics in arid regions.
Response: Thanks you for you recommend. We divided it into two parts, one added to the introduction and the rest to the conclusions
The part of added to the introduction: Vegetation is one of the most important elements of floodplains in arid and semi-arid regions [50-51], and its response to floodplain hydrogeomorphology has attracted increasing attention in recent decades [7-9]. However, methods for quantifying differences in floodplain vegetation responses to changes in hydrogeomorphology from the river centerline to the edges of floodplains in arid regions using MODIS-NDVI time series have received little study. In this study, the study area was divided into four lateral hydro-morphological belts, based on the seasonal hydrological dynamics of the floodplain. Quantitative information on the responses of floodplain vegetation to hydrogeomorphology from the riverbed to interior of the floodplains was obtained, and the hydro-geomorphology changing patterns that influence the distribution and growth of floodplain vegetation were revealed.
The part of added to the conclusions: This study enhance our understanding of the interactions between the vegetation and hydro-geomorphology of large-scale floodplains in arid and semi-arid regions[42].
- You have to state this before presenting the results. Please also provide some details why you use such a low resolution if Landsat images have a 30-m resolution.
Response: We would like to thank the referees for their helpful comments. The advantages and disadvantages of Landsat images in extraction of the vegetation greenness are described. Related information is contained in our revised manuscript. (lines 50-58, page 2):
Readily available Landsat imagery with a long record provides sufficient spatial resolution to capture subtle dynamics of vegetation at the landscape scale but its coarse time resolution and interference of clouds makes acquisition of a detailed time-series of vegetation greenness over a short time period unlikely(Haas et al., 2009). The two sentinel 2 satellites that launched in recent years, like Landsat, can be obtained free of charge all over the world, and also has a high spatial resolution (10-60m). Although the spatial resolution is highly consistent with Landsat, but the revisit time is reduced to 5 days. MODIS(Moderate Resolution Imaging Spectroradiometer) images which 1-day frequency can reflect the change of vegetation greenness in highly dynamic river channel, but their coarse spatial resolution provides a sub-optimal representation of the instantaneous distribution of vegetation greenness, especially in flat landscapes with complex channel networks (Aires et al., 2014; Huang et al., 2013; Justice et al., 1998; Ogilvie et al., 2015; Sakamoto et al., 2007).
- Did you perform any study on this? How can we conclude it from the presented results?
Response: Thanks for the Reviewer’s thoughtful comment. Your comment has greatly helped to improve the paper quality.
The detailed revision can be found on lines 447-453, Page 16:
Floodplain vegetation had obvious spatial zonal distribution characteristics from the riverbed to the edge of floodplains, vegetation growth condition in the regularly inundated floodplain belt is better than that in other belts, and the average maximum NDVI value is 0.23, which is significantly higher than that in other belts, indicating that vegetation growth and biomass were mainly controlled by floodplain hydrogeomorphology. There is a significant correlation between the water persistence times and peak NDVI values, the correlation coefficient is 0.84, at the significance level of 0.05 showed that regular water inundation is better than high-frequency inundation or extremely rare inundation for floodplain vegetation growth.
- Please follow the Journal guidelines for the References' style.
Response: Thanks you so much. The format of the references were updated follow the journal guidelines.
- Suggested references:
Response: Thanks for your suggestion. The references your recommended have been added to the corresponding contents of the paper.
Thank you and the reviewer 2 for his/her valuable suggestions and comments on our work. We look forward to hearing from you regarding our submission, and we would be glad to respond to any further questions and comments that you may have.
Sincerely yours,
Xarapat Ablat
